# Relevance of Autophagy and Mitophagy Dynamics and Markers in Neurodegenerative Diseases

**DOI:** 10.3390/biomedicines9020149

**Published:** 2021-02-04

**Authors:** Carlotta Giorgi, Esmaa Bouhamida, Alberto Danese, Maurizio Previati, Paolo Pinton, Simone Patergnani

**Affiliations:** 1Laboratory for Technologies of Advanced Therapies, Department of Medical Sciences, University of Ferrara, 44121 Ferrara, Italy; carlotta.giorgi@unife.it (C.G.); esmaa.bouhamida@unife.it (E.B.); alberto.danese@unife.it (A.D.); paolo.pinton@unife.it (P.P.); 2Surgery and Experimental Medicine, Section of Human Anatomy and Histology, Laboratory for Technologies of Advanced Therapies (LTTA), Department of Morphology, University of Ferrara, 44121 Ferrara, Italy; maurizio.previati@unife.it

**Keywords:** autophagy, mitophagy, neurodegeneration, multiple sclerosis, Alzheimer’s disease, Parkinson’s disease, biomarker, therapy

## Abstract

During the past few decades, considerable efforts have been made to discover and validate new molecular mechanisms and biomarkers of neurodegenerative diseases. Recent discoveries have demonstrated how autophagy and its specialized form mitophagy are extensively associated with the development, maintenance, and progression of several neurodegenerative diseases. These mechanisms play a pivotal role in the homeostasis of neural cells and are responsible for the clearance of intracellular aggregates and misfolded proteins and the turnover of organelles, in particular, mitochondria. In this review, we summarize recent advances describing the importance of autophagy and mitophagy in neurodegenerative diseases, with particular attention given to multiple sclerosis, Parkinson’s disease, and Alzheimer’s disease. We also review how elements involved in autophagy and mitophagy may represent potential biomarkers for these common neurodegenerative diseases. Finally, we examine the possibility that the modulation of autophagic and mitophagic mechanisms may be an innovative strategy for overcoming neurodegenerative conditions. A deeper knowledge of autophagic and mitophagic mechanisms could facilitate diagnosis and prognostication as well as accelerate the development of therapeutic strategies for neurodegenerative diseases.

## 1. Introduction

Neurodegenerative disorders refer to a large group of pathological conditions in which components of the nervous system lose their structure and function. These diseases are primarily classified according to the clinical features (dementia, tremor, rigidity, and bradykinesia) but especially according to the anatomic distribution of the neurodegenerative lesions [1]. The aggregation of misfolded brain proteins represents the main cause of neuronal damage in hereditary and sporadic neurodegenerative disorders (Figure 1) [2]. Alzheimer’s disease (AD) is a late-onset form of dementia and is considered the most frequent type of neurodegeneration worldwide: nearly 50 million people live with AD and related dementia. AD is characterized by a progressive loss of neurons determined by the aberrant accumulation of tau protein and beta-amyloid protein (Aβ protein) [3]. Human brains express six isoforms of tau, whose cellular function is to stabilize the interactions of microtubules with other proteins. To exert this function, tau protein must be phosphorylated. However, in AD, tau is hyperphosphorylated, a condition that induces conformational changes and the aggregation of the tau protein [3]. The second most common neurodegenerative disease is Parkinson’s disease (PD). Persons affected by this disease present some common symptoms (such as anxiety, depression, rigidity, and tremor) that worsen over time [4]. PD belongs to a class of neurodegenerative diseases named “synucleinopathies”, since the protein aggregations typical of the disease, Lewy bodies (LBs), are formed by different types of proteins, and α-synuclein (α-syn) is the major constituent. Several studies have demonstrated that the aggregation and oligomerization of α-syn are determined by alternative splicing events and by post-translation modifications, such as ubiquitination, oxidative nitration, truncation, and phosphorylation [5]. Furthermore, genetic studies on the familial form of PD have unveiled that specific mutations in the encoding gene, SNCA, accelerate the production of insoluble aggregates and oligomers [4]. Neuronal damage, neuronal loss, and a reduction in brain volume are also important factors inducing long-term disability in patients affected by multiple sclerosis (MS), a complex and multifactorial disorder leading to severe physical or cognitive disabilities and neurological defects [6]. Unlike other types of neurodegenerative disorders, MS is not due to the excessive accumulation of misfolded proteins but is the result of a state of persistent inflammation and adverse immune-mediated processes that activates a cascade of molecular events that provoke demyelination in the white as well as gray matter and subsequent axonal and neuronal damage [7]. Despite the existence of fundamental differences among these neurodegenerative conditions, a growing body of evidence demonstrates that they share important pathogenic mechanisms, of which mitochondrial (dys)function, inflammation, infection, and immune responses are the most frequent [8,9,10]. In addition, in recent years, the autophagy process has also been found to be particularly associated with neurodegeneration [11]. Autophagy is a cellular catabolic pathway in which cytosolic components, bacteria, viruses, macromolecules, and whole organelles are transported to lysosomes for degradation. To exert these multiple functions, specialized forms of autophagy also exist, the most studied being mitophagy (the selective removal of damaged mitochondria) [12]. Autophagy itself and its specialized forms play important roles in physiological as well as pathological conditions. Under normal conditions, autophagy removes unnecessary material, regulates the physiological turnover of organelles, and meets energetic demand. In pathological conditions, autophagy may have both favorable and deleterious roles. As demonstrated, a loss/gain of function in the autophagic process and increase/decrease in the expression of crucial autophagic mediators have been associated with diverse human disorders, particularly cancer [13,14] and neurodegeneration [15]. Most importantly, different studies have not only confirmed the importance of autophagic dynamics during neurodegeneration but also suggested that several proteins involved in this catabolic process may be considered potential markers for predicting neurodegenerative conditions [16,17]. Analysis of the distribution of autophagy partners and regulators along the different pathologic steps of neurodegenerative disorders may improve the knowledge of the contribution of autophagic processes to neurodegenerative conditions. In addition, we may develop innovative neuroprotective therapies and unveil new potential biomarkers for the early diagnosis and clinical management of these diseases. In this review, we discuss the roles of autophagy and its specialized form mitophagy in different neurological disorders. We explore the possibility of using molecular partners of autophagy and mitophagy as biomarkers for neurodegenerative disease status. Finally, the pharmacological modulation of these processes is discussed as a potential strategy for building new therapeutic approaches against neurodegeneration.

## 2. A General Overview of Autophagy

The word autophagy was introduced in late 1963 by the biochemist Christian de Duve [18] and defines a self-degradative cellular pathway whose intent is to degrade and recycle cellular contents. Autophagy exists in three forms that are classified according to their mechanisms and cellular functions: macroautophagy, microautophagy, and chaperone-mediated autophagy (CMA). During microautophagy, the cytosolic material is wrapped and transported directly into the lumen of lysosomes. The main function of microautophagy (mA) is to control cell survival and organellar turnover upon nitrogen restriction. Unfortunately, due to the lack of specific methods for measuring mA (apart from electron microscopy), the effective contributions of mA in mammalian cells remain little studied, and most studies about mA molecular processes are carried out in yeast [19]. Despite this, different investigations suggest that the molecular dynamics of mA existing in yeast may be conserved in mammalian mA. Consistently with this, it has been demonstrated that the endosomal sorting complex required for transport (ESCRT) system is involved in mammalian [20] and yeast mA [21]. Furthermore, a prolonged starvation condition [22,23] as well as cellular treatments with the macrolide compound rapamycin activates mA in both mammalian and yeast cells [21,24,25].

CMA has an important role in protein quality control (QC) and is responsible for degrading a specific subset of oxidized and damaged proteins. The selectivity of CMA is conferred by the existence of a specific pentapeptide motif (KFERQ), which is present in the amino acid sequences of all CMA substrates. This motif is identified by the cytosolic chaperone heat shock-cognate protein of 70 kDa (hSC70), which brings the protein target directly to the lysosome surface [26]. In the last decade, several advances have been made in understanding the molecular mechanisms of CMA. These findings suggest an important contribution of CMA to diverse human diseases, including neurodegeneration [26]. Undoubtedly, the best-characterized and most prevalent form of autophagy in mammalian cells is macroautophagy (hereafter referred to as autophagy), whose multistep process and contribution to the pathophysiology of diverse neurodegenerative conditions will be discussed throughout this review.

Autophagy, a complex intracellular process that is very ancient and has been strongly conserved during evolution, exists to identify and capture a wide group of intracellular components, ranging from low-dimensional biological macromolecules to whole organelles, and bring them to the lysosomal compartment. Its physiological value rests on two main activities. On the one hand, autophagy acts as a QC mechanism that reshapes the cell, ensuring the removal of damaged proteins and organelles [27]. Selective forms of autophagy can specifically target mitochondria (mitophagy), the endoplasmic reticulum (ER; reticulophagy), peroxisomes (pexophagy), and lipid droplets (lipophagy). In addition, autophagy participates in the struggle against invading pathogens (xenophagy), inducing cell defense [12].

On the other hand, lysosomal degradation represents an important source of amino acids and lipids for the de novo synthesis of proteins and lipids. This is of particular importance during starvation, which limits amino acid availability. The limited availability of amino acids affects protein synthesis, which can be performed only in the presence of all the necessary building blocks, in particular, essential amino acids. Under shortage conditions, amino acid pool completeness can be fulfilled only through the degradation of cellular proteins. In such a way, autophagy represents a fundamental survival mechanism, particularly during stress conditions originating from hypoxia or pathogen invasion [27].

Thus, it is not surprising that energy availability can regulate or trigger autophagy and, in particular, that a large number of stimuli converge on metabolic energy sensors, such as mammalian target of rapamycin (mTOR) and 5′ adenosine monophosphate-activated protein kinase (AMPK), which, in turn, regulate autophagy [28].

In cells, mTOR exists in two complexes, mTORC1 and mTORC2, which not only are composed of different protein-binding partners but also regulate different pathways. The primary role of mTORC2 is to regulate cell survival and cytoskeletal organization, while its role in autophagy remains poorly understood. Recent work has shed light on this obscure point. Indeed, the transforming growth factor beta (TGFB)/INHB/activin signaling pathway has been recently identified as an upstream regulator of mTORC2. TGFB-INHB/activin mediates mTORC2 inhibition and regulates the autophagic flux and the cardiac functions in a *Drosophila* cardiac-specific knockdown of TGFB-INHB/activin model [29]. Another investigation recently confirmed the importance of mTORC2 for autophagy. In this case, it has been demonstrated that mTORC2 exists on a molecular axis with the serum- and glucocorticoid-inducible kinase 1 (SGK-1) and, in this state, controls autophagy and mitophagy induction. Consistently, mTORC2- or SGK-1 deficient *C. elegans* models present a perturbed mitochondrial homeostasis and aberrant ROS production, which trigger autophagy and mitophagy. Excessive autophagic and mitophagic fluxes, in turn, result in developmental and reproductive deficits in mTORC2- or SGK-1-deficient animals [30]. Oppositely, the primary role of mTORC1 is to play a pivotal role in cellular catabolic pathways, particularly autophagy [31]. To exert its function, mTORC1 integrates different stimuli, including hormonal stimulation, nutrient availability, and the oxygen level. In the presence of normal levels of energy and amino acids, mTOR inhibits autophagy through specific unc-51-like autophagy-activating kinase 1 (ULK1) serine phosphorylation at the phosphorylation site Ser 757. By contrast, in response to nutritional deprivation, oxygen unavailability, and mitochondrial dysfunction, AMPK activates autophagy through the phosphorylation of ULK1 at Ser 317 and Ser 777 [32]. Interestingly, another research group demonstrated that AMPK may phosphorylate ULK in additional sites. Indeed, by employing a bioinformatic approach, it has been found that ULK1 contains a further four potential AMPK sites [33]. Three of them (Ser 555, Ser 637, and Thr 574) were also identified by mass spectrometry in cells pretreated with an AMPK activator, while the site Ser 467 was confirmed by immunoblotting with phosphospecific antibodies [33]. Unfortunately, this work lacks an analysis of the effect of the different phosphorylations on autophagy. By using SILAC (stable isotope labeling with amino acids) technology, other work mapped 13 new phosphorylation sites of ULK1 [34]. All of them were dependent on nutrient availability, but only Ser 638 and Ser 758 displayed the most significant changes. In addition, time course experiments investigating the response to nutrient availability demonstrated that these phosphorylations were differentially regulated and that mTOR mediated both phosphorylations. Intriguingly, the authors also demonstrated that the phosphorylation at Ser 638 was also mediated by AMPK [34]. Altogether, these findings demonstrate that ULK1 is the key regulator of autophagy, and the occurrence of different protein phosphorylation events is crucial for regulating its activity. Furthermore, the concurrent existence of at least two opposite regulatory pathways that converge on ULK1 signaling (mediated by MTOR and AMPK) allows the cell to better adapt to extracellular and intracellular variations but also affects several pathological conditions.

In the cells, ULK1 forms a complex with autophagy-related (ATG) 13/200-kDa focal adhesion kinase family-interacting protein (FIP200) and ATG101. As reported above, ULK1 activity is mainly regulated by phosphorylation/desphosphorylation events mediated by AMPK and mTOR. In addition, it has been demonstrated that ULK1 is able to phosphorylate itself at Thr 180 [35] and FIP200, ATG13, and ATG101 [36,37] and that the phosphorylation events are regulated by protein phosphatase. Protein phosphatase 2A (PP2A) and protein phosphatase 1D magnesium-dependent delta isoform (PPM1D) regulate the ULK1 phosphorylation [38,39]. PP2C phosphatases (Ptc2 and Ptc3) mediate the dephosphorylation of ATG13 30655342. The ULK1/ATG13/FIP200/ATG101 molecular axis represents the most upstream regulatory complex related to double-membrane vacuole (autophagosome) formation [28]. Autophagosomes symbolize the starting moment of the whole autophagic process, which begins with the formation of double-membrane lined vesicles that fuse together to engulf portions of the cytoplasm. The resulting double-membrane vacuoles are autophagosomes, which can fuse with vesicles of the endocytic pathway at different stages of maturation or directly with lysosomes, becoming autolysosomes. In autolysosomes, acidic hydrolases break down macromolecules into smaller constituents that are released back to the cytosol by lysosomal transporters and permeases. Once activated, the ULK1/ATG13/FIP200/ATG101 molecular axis also phosphorylates and activates coiled-coil, moesin-like BCL2 interacting protein (BECN1) [40,41]. BECN1 can be part of a complex including class III phosphatidylinositol 3-kinase (PI3K) and its regulatory proteins vacuolar protein sorting 34 (Vsp34), p150, and ATG14L. Upon activation, this complex is involved in the nucleation and elongation of autophagosomes. The first step occurs on the surface of the membranes of the ER, mitochondria, Golgi complex, endosomes, or plasma membrane [42] and consists of the phosphorylation of phosphatidylinositol to form phosphatidylinositol-3-phosphate (PI3P). This phosphoinositide behaves as a positive regulator of autophagy. In fact, the presence of PI3P at the source membrane triggers the docking of several adaptor proteins, which, in turn, induce and sustain the elongation of the sack-like, omega-shaped structure, which grows, binds, and surrounds the material intended to be digested.

Another interaction of BECN1 can exert an inhibitory effect on autophagy [43]. BECN1 has been reported to bind B-cell lymphoma (BCL)-2, BCL-XL, and other members of the BCL-2 family through the BCL-2-homology-3 (BH3) domain. The consequence of this interaction is a diminution of the interaction between BECN1 and the class III PI3K complex, which prevents the formation of phagophores [43]. Accordingly, BCL-2 phosphorylation can reverse BECN1 sequestration and restore autophagy stimulation [43].

The other two systems, ATG12–ATG5–ATG16L1 and microtubule-associated protein 1A/1B-light chain 3 (LC3)–phosphatidylethanolamine (PE) complexes, seem to play an important role in the elongation and closure of autophagosomes, although the underlying mechanism has not yet been clarified. A key process during autophagosome elongation and closure is the lipidation of the LC3 protein, which is joined to the membrane PE. Once inserted into the autophagosomal membrane, the lipidated complex can further recruit other adaptor proteins. This allows autophagosomes to recognize cargo material, and elongate and close the vesicle. The fusion of the autophagosomes with the lysosome is the subsequent step, which, in a normally operating lysosome, is followed by lysosomal compartment acidification, the degradation of macromolecules by hydrolases and lipases, and the recycling of the base constituents (Figure 2).

## 3. Mitophagy: The Master Regulator of the Mitochondrial Population

Mitochondria are essential intracellular organelles that supply substrates and energy to execute numerous cell functions, such as metabolism, differentiation, apoptosis, cell movement, and differentiation. In contrast to other intracellular components, mitochondria are constituted by two membranes, the outer mitochondrial membrane (OMM) and the inner mitochondrial membrane (IMM), which fully surround the mitochondrial matrix. Between the OMM and IMM, another mitochondrial subcompartment exists, the intermembrane space (IMS) [44]. Another unique feature of mitochondria is that they have their own genome (mitochondrial DNA, mtDNA), which encodes 13 proteins that are essential components of the oxidative phosphorylation (OXPHOS) system, the process by which ATP is formed [45]. A series of members (complexes I-V, C-I-V) of the mitochondrial electron chain (mETC) found in the IMM permit the transfer of electrons from NADH or FADH_2_ to O_2_ [46]. The energy produced during this movement creates a proton gradient that is used by the last component of the mETC (C-V, ATP synthase) to synthesize ATP [46]. The impairment of electron transfer or stress conditions affect the production of reactive oxygen species (ROS), of which C-I and C-III are the main producers [47]. Mitochondria are also central hubs for calcium (Ca^2+^) signaling [48]. At rest, mitochondria have low Ca^2+^ concentrations [Ca^2+^] (~100 nM range or lower). However, upon stimulation, mitochondrial [Ca^2+^] can increase to the range of hundreds in micromolar concentration [49]. This happens due to the highly specialized contact sites (mitochondria-associated membranes, MAMs) that exist between mitochondria and the main intracellular Ca^2+^ store of cells, the ER [50]. These interaction sites represent critical hubs for the regulation of diverse cellular processes (such as energy metabolism, inflammation, redox regulation, and lipid and protein transfer), and recently, MAMs have been described to play an important role in the onset and progression of several human diseases by regulating Ca^2+^ transmission between the ER and mitochondria [51]. Once released from the ER, Ca^2+^ can enter mitochondria owing to the close proximity of the ER to mitochondria, the electrochemical driving force (mitochondrial membrane potential) that is created by electron transfer, and the activity of the components of the mitochondrial Ca^2+^ uniporter (MCU) [52,53]. Mitochondria are normally present in cells in the form of a dynamic network, where the mitochondrial mass increases as a consequence of mitochondrial biogenesis. The control and reshaping of the mitochondrial population can occur through different mechanisms [54]. These mechanisms include (i) the control of protein quality through mitochondrial proteases, the mitochondrial unfolded protein response, or proteasome-dependent degradation; (ii) the budding of mitochondrion-derived vesicles; and (iii) the targeting of some or all mitochondria to lysosomes through mitophagy.

Mitophagy regulation is not yet a completely understood process. During short-term starvation, the mitochondrial pool is not depleted, so as to not further reduce the cellular production of energy, while oxidative metabolism is mainly sustained by general autophagy [28]. This fact necessarily implies a difference in regulation between autophagy and mitophagy that allows the cautious sparing of mitochondria, which are among the principal end-users of the material provided by autophagy. A role in this sense seems to be played by fission restriction. In fact, fragmented mitochondria appear to be a preferred target for mitophagy: when their number is reduced, mitophagy itself is restricted.

When the ultimate goal is to eliminate mitochondria, there are different physiological mechanisms that can be activated. The first example is programmed mitophagy. There are several situations in the cell that can require the activation of programmed mitophagy, independent of the wellness of mitochondria. An example is the mitochondrial depletion that occurs in reticulocytes during differentiation through the activity of NIP3-like protein X (NIX/BNIP3L). Other examples include the elimination of male-derived mitochondria after egg fertilization [55] and the reshaping of the mitochondrial population during cardiomyocyte [56] or muscle cell differentiation, which induces a change from carbohydrate- to fatty acid-driven OXPHOS [57]. Stimulations that can normally trigger mitophagy can be affected by mitochondrial defects, such as a decline in transmembrane potential and excessive ROS production.

Mitophagy involves some fundamental steps. First, as stated above, mitochondria must assume the dimensions necessary to easily enter autophagosome vesicles. Therefore, they are normally resized through fission processes. In addition, they need to be properly displayed on the surface to trigger the formation of vesicles, which will engulf them. Typically, “eat-me signals” can be ubiquitin-dependent or not. The best-known example of a ubiquitin-dependent mechanism is the PTEN-induced kinase 1 (PINK1)/Parkin axis. PINK1 and Parkin belong to a series of genes referred to as PARK genes, which include α-syn (PARK1/4), Parkin (PARK2), PINK1 (PARK6), protein deglycase-1 (DJ-1, PARK7), leucine-rich repeat kinase 2 (LRRK2, PARK8), and ATP13A2 (PARK9). The name of this group of genes (Parkin genes) comes from the finding that mutations in these genes have been linked to familiar forms of PD. In particular, approximately 100 mutations in the Parkin gene have been identified as causing autosomal recessive Parkinsonism [58].

PINK1 is a mitochondrial serine/threonine-protein kinase, and Parkin is an E3 ubiquitin ligase; these proteins induce different functions at the cellular level but act in a common pathway to regulate mitophagy.

PINK1 is a ubiquitous protein characterized by a mitochondrial targeting sequence (MTS), a transmembrane domain, and a highly conserved serine/threonine kinase domain. At present, approximately 30 pathogenic PINK1 mutations that impair its kinase activity and provoke loss of function have been identified [59,60,61,62].

Normally, PINK1 is imported into mitochondria via the activity of the translocase of the inner membrane (TIM)–translocase of the outer membrane (TOM) complex. Once PINK1 arrives in the IMM, it is subjected to a series of proteolytic cleavages that reduce the full-length form of PINK1 into fragments, which are then degraded by the proteasome [63,64,65]. In the presence of alterations in mitochondrial membrane potential, the activity of the TIM/TOM complex is reduced, and PINK1 begins to accumulate on the OMM. Here, after being stabilized by a molecular complex including TOM proteins [66,67], PINK1 phosphorylates Parkin. The phosphorylation converts Parkin from an autoinhibited enzyme to an active ubiquitin (Ub)-dependent enzyme [68,69]. In this state, Parkin actively ubiquitinates several mitochondrial proteins at the OMM. The ubiquitination events promote the recruitment of the Ub-binding autophagy receptors p62/Sequestome, NBR1, NDP52, optineurin (OPTN), and TAX1BP1 (TBK1), which connect damaged mitochondria to phagosomes for clearance in lysosomes [70,71,72]. In recent years, different studies have identified pathways regulating mitophagy that are PINK1–Parkin-independent. These mechanisms may act in parallel or in addition to PINK1–Parkin-dependent mitophagy and involve a series of OMM mitophagy receptors that bind LC3 and recruit mitochondria to autophagic vesicles. Among them, the most studied are the proapoptotic members of the BCL2 family, NIX and BNIP3 [73,74] and FUNDC1 [75], which regulate the mitophagy process during ischemic/hypoxic conditions, and the BECN1 regulator AMBRA1. Interestingly, it has been proven that AMBRA1 regulates both Parkin-dependent and Parkin-independent mitophagy [76] (Figure 2).

## 4. Relationship between Autophagy and Mitophagy in MS

Multiple sclerosis (MS) is a progressive and chronic disease that affects approximately 3 million persons worldwide. MS is an inflammatory condition in which activated immune cells enter the central nervous system (CNS) and cause progressive demyelination, gliosis, and neuronal loss. The symptoms vary from individual to individual [77]. The most common symptoms are walking difficulties, sensory disturbances, vision problems, and cognitive and emotional impairments. Typically, MS starts with an unexpected onset of neurological impairments, and the majority of individuals display a relapsing–remitting (RR) course of the disease in which recurrent periods alternate with relapse phases. This course may be followed by a secondary progressive phase in which inflammatory attacks are more frequent and cause irreversible neurological impairments. A small percentage of individuals may present with the primary progressive form of the disease, which is characterized by the absence of remission periods and a progressive worsening of symptoms [78]. Currently, the pathogenesis and etiology of MS are unclear. MS is considered a multifactorial disease, and genetic predisposition and environmental factors may play important roles in disease progression. Furthermore, mitochondrial dysfunction as well as the impairment of the QC systems of mitochondria have been identified in different MS samples and represent evidence that the mitochondrial compartment has a major role in MS [9]. In addition, recent investigations have described an important contribution of autophagic processes. The first evidence that autophagy could be involved in MS was reported in 2009, when a strong correlation was found between the expression of the autophagic marker ATG5 and the clinical disability observed in the experimental autoimmune encephalomyelitis (EAE) MS animal model. Moreover, in this work, the authors found increased expression of ATG5 in T cells obtained from RR-MS patients and in postmortem brain tissue from individuals with secondary progressive MS [79]. Unfortunately, the authors did not address the role of autophagy in T cells and MS. They only speculated that autophagy may help to increase the survival of T cells and help to propagate the immune response. Similarly, other work detected ATG5 increases in terms of both mRNA levels and protein amounts in T cells obtained from MS patients who were treatment naïve [80]. Increases in ATG5 also correlated with the presence of proinflammatory cytokines, thus displaying a possible relationship between the inflammatory status and ATG5 expression in MS. However, they did not perform a detailed analysis of the clinical activity state [80]. T cells present different subpopulations. Among them, T regulatory cells (Treg) are particularly relevant in autoimmune disease because they prevent inflammation and preserve the tolerance to self-antigens. Recently, it has been demonstrated that the autophagic mediator AMBRA1 associates with the protein phosphatase PP2A to sustain Treg differentiation by increasing the expression of Forkhead box P 3 (FOXP3), an essential transcription factor for the differentiation of Treg cells [81]. In addition, the AMBRA1–PP2A–FOXP3 molecular axis was found to be essential for regulating the optimal autophagic levels necessary for T-cell stimulation and differentiation. Consistently, AMBRA1 conditional KO mice display reductions in FOXP3 levels with consequent impairments in Treg differentiation and activity. Most importantly, AMBRA1 deficiency worsens the disease pathogenesis in an EAE MS animal model [81]. Finally, work of Akatsuka et al. not only demonstrates the important role of AMBRA1 in the regulation of T cells, but also highlights decreased mitochondrial functioning and metabolism in these cells [82]. All these findings demonstrate that AMBRA1 is an essential factor that regulates both autophagic and mitochondrial behaviors and, probably, also the mitophagic process in T cells.

In MS, T-cell activities may be modulated by the complement-regulating molecule CD46 [83]. This factor is also described as an autophagic inducer [84], and its levels are documented to be increased in the serum and cerebrospinal fluid (CSF) of MS patients [85]. The increased T-cell autoreactivity in MS may also be promoted by IRGM1, a GTPase that regulates the survival of immune cells through autophagy. Consistent with this finding, IRGM1 deletion increases the apoptosis of T cells, reduces their proliferative capacity, and ameliorates the clinical score of the EAE mouse model [86]. Considering that subsequent studies have demonstrated that IRGM1 is localized to the mitochondrial compartment and regulates the mitochondrial metabolism and mitochondrial fission induced by mitophagy [87,88], the increased T-cell autoreactivity observed in MS may be due to an impairment in the mitophagic process. In addition to its effects on T cells, autophagy plays a role in dendritic cells (DCs), the most potent antigen-presenting cells (APCs) in the immune system. In particular, autophagy starts in response to bacterial and viral infection. By generating transgenic mice with silencing of ATG7 in DCs, Bhattacharya and colleagues demonstrated the importance of DCs and autophagy in MS. Indeed, they showed that the specific loss of autophagy in DCs significantly delayed disease progression and reduced disease severity in EAE mice [89]. As reported above, AMPK is the main positive regulator of autophagy. This kinase works by sensing the AMP/ATP ratio and activates autophagy to combat energetic imbalance. It has been demonstrated that following exposure to proinflammatory cytokines, AMPK activates and triggers autophagy in oligodendrocyte precursor cells (OPCs) [90]. This change is due to a metabolic switch from OXPHOS to glycolysis and impairment of mitochondrial dynamics, leading to increased oxidative stress and reduced mitochondrial Ca^2+^ uptake and ATP production. As a consequence, OPCs fail to differentiate into mature and myelinating oligodendrocytes [90]. In support of these in vitro findings, recent work demonstrated that metabolic stress-induced autophagy is a key element in an in vivo MS model. Indeed, MCU-deficient (MCU-def) mice subjected to EAE displayed elevated clinical scores, excessive inflammation, and demyelination [91]. Morphological and functional analyses performed with the spinal cords of MCU-def mice revealed important mitochondrial damage, accompanied by an elevated presence of autophagosomal markers and a decrease in ATP synthesis and mitochondrial gene expression. Overall, these data confirm that the presence of mitochondrial dysfunction provokes the inhibition of Ca^2+^ buffering, ATP synthesis, and mitochondrial gene expression, causing a metabolic collapse that prompts autophagy and worsens MS-like conditions. Furthermore, since autophagic activation accompanied by the downregulation of PGC1α (a master regulator of mitochondrial biogenesis) has been observed, it is possible to speculate that the mitochondrial QC system is also affected. However, studies have not verified whether autophagy activities lead to autophagic mitochondrial removal.

Markers of autophagic processes may represent reliable potential biomarkers for monitoring the progression of disease. Increased amounts of Parkin, ATG5, and inflammatory cytokines are present in both the serum and CSF obtained from MS patients. Analyses comparing MS patients to healthy individuals and patients affected by other neurodegenerative conditions have been conducted [16]. Moreover, subsequent work demonstrated that increases in both autophagic and mitophagic markers correlated with the active phases of the disease and with circulating lactate levels, demonstrating the presence of an impaired metabolic status in MS patients [92]. Notably, several studies have associated lactate levels with MS progression [93]. Other independent research groups have confirmed that circulating autophagy and mitophagy markers are increased in MS biofluids [94,95]. In addition, the circulating levels of mitochondrial adenine nucleotide translocase 1 (ANT1) and oxidative stress markers have also been investigated. Interestingly, MS patients display increased oxidative stress, accompanied by reduced levels of the mitochondrial marker ANT1, suggesting that the mitochondrial QC systems are activated to promote the removal of nonfunctioning mitochondria. Consistent with this, reduced circulating levels of the OMM protein translocator protein 18 kDa (TSPO) and increased amounts of the mitochondrial disease marker growth/differentiation factor 15 (GDF-15) have been found in MS individuals and correlate with the severity of the disease [96,97] (Table 1). 

It is clear that autophagy and mitophagy as well as the mitochondrial quality control system are important contributors in MS. In the last few years, an increasing number of studies have correlated the activities of such molecular mechanisms with the progression of the disease. Furthermore, circulating elements of autophagy and mitophagy may be detected in human samples from MS individuals, thus suggesting the possibility of using them as novel biomarkers. However, MS shows a great heterogeneity with regard to the clinical symptoms as well as therapy response. In addition, MS manifests in different forms (clinically isolated syndrome, RR MS, secondary progressive MS, and primary progressive MS), where the relapse rate and disability progression differentiate one from the other. Only when the dynamics and response of autophagy and mitophagy are well characterized in regard to all these conditions will we be able to claim to have identified the real contributions of them in MS, and we could use autophagic and mitophagic elements as innovative markers for MS disease progression.

## 5. Involvement of Autophagy Mechanisms in AD Progression

AD was first described in the early 20th century and is characterized by a progressive deterioration of cognitive function. Memory loss and dementia represent the most common symptoms. The cardinal pathological hallmarks of AD are extracellular (amyloid) plaques and intracellular and extracellular neurofibrillary tangles (NFTs). Amyloid plaques are composed of deposits of Aβ, α-syn, Ub, and apolipoprotein E. NFTs are characterized by hyperphosphorylated tau protein and apolipoprotein E. These aggregates induce neuronal toxicity by impeding neural communication and provoking cell death either directly or by preventing the delivery of an optimal nutrient supply to brain cells [3].

At present, the origin of AD and the mechanisms occurring in the pathogenesis of AD are not well defined. Inflammation seems to play an important role: mediators of inflammation, such as cytokines, adhesion molecules, and prostaglandins, drive degeneration in different neural AD models [98]. Consistent with this finding, aggregated peptides increase proinflammatory agent production, and inflammatory molecules are detected in the CSF, serum, and plaques obtained from AD patients. Oxidative and nitrosylative damage provoked by ROS and reactive nitrogen species (RNS) are determinants of the initiation and progression of AD [99]. Oxidatively damaged membrane phospholipids and increased oxidative stress in neurons are frequently present in neurons exposed to Aβ [100]. Furthermore, AD brains extracted at autopsy have decreased amounts of vitamins A and E and β-carotene [101] and display a higher production of free radicals and increased expression of neuronal nitric oxide synthase (nNOS) [102]. This increased nNOS correlates with an increased apoptosis of hippocampal neurons. In the last 10 years, an increasing number of studies have demonstrated the critical contributions of autophagy and mitophagy to AD pathogenesis [103]. Several studies have reported an increased presence of Aβ in autophagosomes [104]. Interestingly, autophagosomes also contain amyloid precursor protein (APP) and its processing enzymes, in particular, a component of the γ-secretase complex, suggesting an additional source of Aβ. Consistent with this finding, the induction of autophagy correlates with Aβ production, and autophagy-deficient animals (with ATG7 knockdown) display reduced Aβ secretion [105]. Additionally, the hyperphosphorylation of tau correlates with increased autophagic levels. Indeed, postmortem AD brain samples are characterized by LC3- and p62-positive autophagosomes, and the hyperphosphorylation of tau has been recognized in autophagy-deficient mice [106,107]. Although these observations highlight a dangerous correlation between autophagy and AD, other studies suggest that autophagy and mitophagy may exert beneficial effects against AD [103]. The abnormal accumulation of autophagosome vesicles is present in AD neurons [104]. This accumulation is related to compromised lysosomal function, which results in lysosomes that are no longer able to degrade autophagosomes. The overexpression of Parkin and PINK activates mitophagy, restores mitochondrial function, and reduces Aβ production [108,109]. Similar results have been obtained from another independent experiments that demonstrated that mitophagy is essential for reducing Aβ levels, abolishing tau hyperphosphorylation, preventing cognitive impairments in an AD mouse model, and suppressing neuroinflammation [110].

To confirm the crucial role of autophagic and mitophagic dynamics in AD, different studies have evaluated the presence of elements belonging to these processes in biofluids from persons with AD. The first investigation was performed in 1995, in which ventricular CSF from postmortem AD patients was analyzed. In this study, the authors detected increased levels of the lysosomal protein cathepsin D [111]. However, a subsequent report performed with lumbar CSF samples from living AD patients found no change in the levels of cathepsin B [112]. This finding was confirmed in other work that investigated a broad range of lysosomal proteins in CSF samples from living AD patients and found no variations in diverse cathepsin forms (A, B, D, and L); however, the study did find altered expression for five other lysosomal proteins in the AD samples: early endosomal antigen 1 (EEA1), LAMP1, LAMP2, RAB3, and RAB7 [113]. By contrast, a recent study analyzed the levels of proteins associated with lysosomal function in the CSF of AD persons by conducting solid-phase extraction and parallel reaction monitoring mass spectrometry and found only minor or absent changes in their levels [114]. Unfortunately, the levels of proteins directly related to autophagy and mitophagy processes were not investigated in that study. A follow-up study at 12 and 24 months identified autophagic elements (BECN1, p62, and LC3) in peripheral blood mononuclear cells (PBMCs) obtained from the blood of AD patients and demonstrated that their levels varied during the course of the disease and correlated with the inflammatory environment [115]. Recently, autophagic elements have also been assessed directly in AD blood samples. Indeed, increased levels of the autophagic marker ATG5 are present in the plasma of patients with dementia who meet the criteria for probable AD. Unfortunately, the authors did not identify the subtype of dementia or confirm the AD status. These limitations were overcome in a recent investigation assessing the circulation of autophagic and mitophagic markers in the serum of patients affected by mild–moderate late-onset AD, mild cognitive impairment (MCI), vascular dementia (VAD), and mixed dementia (MD). In this work, the authors found decreased levels of ATG5 and Parkin in patients affected by AD, MCI, and MD. By contrast, they detected increased levels of these markers in VAD patients [17]. This investigation suggests that autophagy and mitophagy markers are possible biomarkers for AD and that they are differentially affected in different dementia types, which may help to discriminate AD-type dementias from VAD. Additionally, the fact that AD samples have decreased levels of autophagy and mitophagy markers confirms the presence of an impaired degradative system in AD persons (Table 1).

Summing up, autophagy and mitophagy represent well-established mechanisms in AD and may exert a protective role. Accordingly, most research highlights the reduced recruitment of both autophagy and mitophagic factors in cell cultures, in vivo AD models, and human samples obtained from AD-affected patients, including in the body fluids of the CSF and blood. Here, autophagic and mitophagic partners also correlate with the inflammatory status and change during the course of the disease, thus opening up the possibility of using autophagic and mitophagic elements as markers for the progression of AD. However, before ascribing merit to these molecules as potential screening, prognostic, diagnostic, or disease-monitoring markers for AD, it is important to consider different aspects. The diagnosis of AD cannot be achieved until the patient displays dementia symptoms. In addition, different dementia types exist and vary between individuals. Very few studies have monitored the variation of circulating markers of autophagy and mitophagy during the different dementia types. Furthermore, these studies lack validation of the investigated markers with accepted methods for diagnosing AD, such as amyloid PET imaging. Again, all the investigations performed did not provide follow-up studies and did not analyze the effects of the disease-modifying drugs commonly used for AD therapy on autophagy and mitophagy circulating markers. Undoubtedly, more detailed analyses and larger cohort studies are necessary to verify whether autophagic and mitophagic circulating elements may represent promising biomarkers for AD.

## 6. Current Knowledge of the Relationship between PD and Autophagy Dynamics

Resting tremor, bradykinesia, rigidity, and postural instability represent the four cardinal signs of PD, the most common neurological movement disease, and PD is characterized by a progressive loss of dopaminergic neurons in the substantia nigra pars compacta. PD is considered a multifactorial disease since both genetic and environmental factors play important roles. Approximately 90% of the cases are sporadic, while the remaining 10% are caused by monogenic mutations in at least 23 genes. Similarly, a number of cellular mechanisms are involved in PD pathogenesis. Among them, the uncontrolled intracellular aggregation of α-syn, in the form of LBs and Lewy neurites, represents the main hallmark of the disease. It is not surprising that the first evidence of the genetic mechanisms of PD was a mutation in the α-syn gene, and different forms of the α-syn protein (oligomers, protofibrils, and unfolded monomers) have been found in human PD brain samples. α-syn is a presynaptic neuron protein abundantly expressed in the nervous system; it is present in proximity to synaptic vesicles and folds into α-helical structures. The primary role of this protein is to attenuate neurotransmitter release and synaptic vesicle recycling. In PD, α-syn generates β-sheet structures that are prone to aggregation, which leads to pathologic conditions with toxic gain-of-function effects. Mitochondrial dysfunction is another crucial element during PD pathogenesis in both sporadic PD and familial Parkinsonism. Different postmortem studies have highlighted the existence of deficiencies in components of the mETC, and compounds (toxins and pesticides) were found to promote the Parkinsonian phenotype and neuron loss by impairing complex I of the mETC. Furthermore, α-syn alone induces effects by interacting with the mitochondrial membrane, accumulating inside the organelle, and leading to mitochondrial dysfunction and oxidative stress by damaging C-I [116]; alternatively, α-syn can interact with the mitochondrial transporter TOM20 [117]. Another gene causing autosomal dominant PD is LRRK2, a protein involved in diverse signaling pathways, including vesicular trafficking, protein translation, and the control of mitochondrial dynamics. Mutations in this member of the leucine-rich kinase family have been found in approximately 1–2% of sporadic and 5% of familial PD cases. The most frequent mutation of LRRK2 (G2019S) induces an increase in LRRK2 activity. G2019S-LRRK2 PD postmortem human tissues, animal models, and cellular models are characterized by important mitochondrial dysfunction, with impaired ATP production, mitochondrial fragmentation, mtDNA damage, and oxidative stress representing the main features. Recently, it has been demonstrated that this mutation induces impairment in mitophagic clearance [118]. In addition, the loss of function, mutation, and overexpression of the mitophagic regulatory members PINK1 and Parkin provoke impaired mitochondrial turnover and cause autosomal recessive PD. To mediate mitophagy, PINK1 and Parkin cooperate to recognize and label damaged mitochondria with polyubiquitin (p-Ub) chains [119]. Postmortem brains from LB disease patients are characterized by p-Ub chain structures that colocalize with markers of mitochondria and autophagy [120]. Mutant forms of PINK1 are unable to move to mitochondria upon stress signaling, thereby avoiding mitophagic induction. Similarly, Parkin mutations (such as S65N, G12R, and R33Q) decrease the capacity of PINK1 to phosphorylate and activate Parkin itself. Correct mitochondrial turnover is not guaranteed by only the mitophagic process; it is also regulated by fission and fusion events. Among the mitochondrial fusion proteins, the best characterized are mitofusins (MFNs). It has been demonstrated that MFN-1 and MFN-2 are substrates for PINK1 and Parkin and that they can be ubiquitinated by both PINK1 and Parkin [121,122]. Consistent with this, mutations in and the loss of PINK1 and Parkin impair MFN-1/2 ubiquitination in PD patient cells [123]. MFNs also work as bridges between mitochondria and the ER to preserve the appropriate functioning of MAMs. These contact sites between the ER and mitochondria act as primary signaling hubs for cells and regulate lipid homeostasis, calcium dynamics, apoptosis, the stress response, and autophagosome vesicle formation. Diverse recent studies have highlighted a contribution of these contact subdomains in the progression of PD [124]. Several proteins encoded by genes involved in PD (α-syn, Parkin, and PINK1) are located in MAMs and have been found to regulate correct ER–mitochondrion tethering. For example, Parkin deletion increases MFN2 amounts and increases Ca^2+^ transfer from the ER to mitochondria [125]. Notably, this event is a crucial mediator of the regulation of autophagy [126]. In addition, DJ-1, which provokes a rare form of autosomal recessive PD [127], increases ER–mitochondrion communication and preserves the optimal Ca^2+^ transfer between the ER and mitochondria, thereby having a cytoprotective role that is essential for maintaining mitochondrial functioning [128] and, probably, autophagy levels. As reported above, alterations in tau protein are highly related to AD disease. In truth, pathogenic mutations in tau protein are present in different neurodegenerative disorders, defined as tauopathies. Interestingly, different studies suggest that tau mutations are also present in PD. By comparing characterized tau mutations related to tau toxicity and aggregation in PD (P301L and A152T) [129,130,131], a recent investigation explored, for the first time, the concomitant activity of the three different forms of autophagy (autophagy, CMA, and mA) [132]. Here, the authors found different activity levels for the three autophagic forms and demonstrated that the pathogenic tau mutation A152T resulted in a blockage of both CMA and mA, but caused a compensatory activation of autophagy. Oppositely, the P301L mutation provoked an inhibition of the degradation of tau aggregates by any of the three catabolic pathways [132]. A deeper understanding of the different recruitment of the diverse autophagic forms may help to increase our knowledge of the molecular mechanisms existing in PD as well as in the other tauopathies.

Detecting autophagy- and mitophagy-related proteins in peripheral human biospecimens may represent a promising method for identifying PD statuses and controlling the progression of the disease. Significant decreases in LC3B, BECN1, ATG5, and lysosomal associated membrane protein (LAMP) 2 are present in CSF samples obtained from early-stage PD patients. Interestingly, among these autophagic partners, only LC3B shows a significant correlation with α-syn, total tau levels, and the clinical severity of patients [133]. Notably, recent studies have demonstrated that tau protein also participates in the pathology of PD [134]. A reduction in LAMP2 levels in the CSF of PD patients was found in another study. In this study, the authors found a decrease in LAMP1 levels, but they did not identify variations in LC3 levels [135]. An important difference exists between these two studies. In the first, circulating proteins were detected by using ELISA technology, while the second employed an immunoblotting technique. LAMP2 levels were also analyzed in a study comparing PD patients, PD patients harboring LRRK2 mutations, and healthy control subjects with or without LRRK2 mutations. The main finding of that investigation was that LAMP2 protein levels were reduced in the PD patients harboring LRRK2 mutations. Similar to the study mentioned above, LAMP2 levels were not related to the clinical states of the patients. However, a positive correlation between LAMP2 and oxidative stress has been shown [136]. Autophagic markers and related proteins were analyzed in circulating PBMCs obtained from PD patients. The results demonstrated that the steady-state autophagy in PD patients was profoundly different from that observed in healthy individuals and correlated with augmented expression of α-syn [137]. Unfortunately, investigations aimed at detecting mitophagic elements in human biofluids are lacking. The existing literature can inform us about the “mitochondrial signature” that exists in PD patients. Significant decreases in mtDNA copy number have been observed in patient blood cells and in CSF samples from early-stage PD [138,139]. Increased mtDNA levels were present in the CSF of PD patients carrying LRRK2 mutations [140] and in the sera of PD persons with mutations in Parkin or PINK1. This study demonstrated an association between mitophagy impairment (represented by a Parkin- or PINK1-mutant genotype) and mtDNA in circulating biofluids from PD patients for the first time. Finally, an interesting study found reduced methylated mtDNA in PD patients, suggesting that affected patients may have disrupted mtDNA gene expression and replication [141] (Table 1).

Undoubtedly, the discovery that many genes involved in both autophagy and (in particular) mitophagy are mutated in familial Parkinsonism and in sporadic PD makes these molecular processes fundamental for this neurodegenerative disease. The fact that markers of these catabolic systems may be detected in human samples also opens up the possibility of creating new real-time monitoring approaches for the progression of the disease. However, today, these studies’ results are still incomplete, and some of them report controversial results, probably due to the limited sizes of the cohorts analyzed, the different types of samples analyzed, and, most importantly, not always accounting for the clinical history of each single patient. It is thus clear that more detailed and larger longitudinal, stratified, and standardized analyses are needed.

## 7. Principles and Current Strategies for Targeting Autophagy in Neurodegeneration

Several therapeutic approaches ameliorate the consequences and symptoms of neurodegenerative disorders. Unfortunately, current treatments become less effective as the neurodegenerative status advances, and most importantly, none of them prevents the onset or progression of the disease. The evidence reported in the previous sections demonstrates that autophagy processes have an important role during the development of the most common neurodegenerative diseases. Hence, these findings suggest that the modulation of autophagic and mitophagic processes may be a possible innovative therapeutic approach for combating neurodegeneration (Figure 3).

Autophagy upregulation has been demonstrated to be an effective strategy for increasing the clearance of neurodegenerative disease-causing proteins in different cellular and mouse models, thereby reducing toxicity. Autophagic activation may be induced by blocking mTOR activities. Rapamycin is the best known mTOR inhibitor, and it has been demonstrated that this compound increases the clearance of tau protein and decreases tau toxicity [142,143]. In addition, rapamycin activates autophagy to remove other protein aggregates, such as long polyglutamines and polyalanine-expanded proteins [143]. Unfortunately, one limit of rapamycin is its limited absorption. Different analogs of rapamycin (rapalogs) have been developed in recent years. Among them, temsirolimus has been found to increase the autophagy clearance of hyperphosphorylated tau and ameliorate learning and memory impairments [144]. Another compound mediating mTOR inhibition is the proneurogenic and antihistaminic compound latrepirdine. It has been demonstrated that latrepirdine also improves learning behaviors and reduces Aβ and α-syn aggregates in an AD mouse model [145]. Autophagy can also be activated by mTOR-independent pharmacological agents. For example, resveratrol exerts neuroprotective effects in PD models by increasing autophagy through an AMPK-dependent mechanism [146]. The same effects on AMPK activity are induced by the recently identified small molecules A769662, GSK621, RSVA314, and RSVA405. A769662 and GSK621 promote the autophagic clearance of α-syn aggregates by inducing the phosphorylation of AMPK and ULK1 [147]. RSVA314 and RSVA405 activate AMPK by a CaMKKβ-dependent mechanism to activate autophagy and promote the degradation of Aβ 20852062. The molecule AUTEN-67, which antagonizes the autophagic inhibitor phosphatase MTMR14, increases autophagic flux, promotes neuron longevity, and prevents neurodegenerative symptoms in AD models [148]. Interestingly, the same research group also demonstrated that another molecule, AUTEN-99, is capable of improving autophagy and exerting neuroprotective effects in PD models [149]. Lithium is widely used to treat bipolar disorders and depression. In addition, recent investigations have demonstrated that lithium administration activates autophagy by inhibiting inositol monophosphatase in an mTOR-independent manner [150] and exerts neuroprotection in AD [151]. Consistent with this, a clinical trial evaluating long-term treatment with lithium in AD patients revealed the amelioration of multiple cognitive parameters in the lithium group. Furthermore, analyses conducted with CSF samples revealed a significant reduction in phosphorylated tau [152]. It is clear that an improvement in autophagic machinery should permit an increase in the clearance of protein aggregates. Despite this understanding, studies have also demonstrated that pharmacological interventions aimed at blocking the autophagic process may be useful for counteracting a neurodegenerative status. In an α-syn transgenic mouse model, the overexpression of α-syn reduced dendritic and synaptic markers, which were reduced after exposure to the anti-autophagic compounds bafilomycin-A1 (Baf-A1) and chloroquine (CQ). Interestingly, the authors also found a reduction in α-syn inclusions after treatment with rapamycin. This finding suggests that the aggregation of α-syn is not exclusively mediated by the mTOR-dependent regulation of autophagy. Considering that mTOR is involved in multiple cellular processes, it is possible that other mechanisms are involved in α-syn metabolism [153]. Of note, it is important to specify that both CQ and Baf-A1 have a broad spectrum of biological activities. Similar to what was observed for α-syn aggregates, the inhibition of autophagy promoted by CQ induced a reduction in total tau levels in rat hippocampal extracts. Increase and decreases in tau levels have been observed after rapamycin administration [154]. The inhibition of autophagy also seems to exert beneficial effects in MS. Indeed, by suppressing the inflammatory process in EAE, CQ administration ameliorates the clinical signs of the disease [155]. A subsequent study unveiled that the effect of CQ in reducing the severity of the clinical course of EAE is mediated by a direct effect on DCs. In this work, the authors demonstrated that CQ-treated cells displayed reduced expression of molecules involved in antigen presentation, which resulted in reduced T-cell activation and proliferation [156]. The deleterious role of autophagy in MS has also been demonstrated in the cuprizone (CPZ) demyelination model, which permits us to determine the contribution of other elements independent of the immune system during demyelination/remyelination. The administration of CPZ with rapamycin resulted in increased demyelination compared with treatment with CPZ alone [157]. Furthermore, rapamycin increases axonal damage and leukocyte infiltration when administered together with CPZ [158], suggesting that by administrating agents blocking the autophagic process, it may be possible to reduce demyelination. Despite this finding, other studies have demonstrated that rapamycin ameliorates histological and clinical signs in MS models, particularly EAE models [159]. In addition, a clinical trial in which RR-MS patients received rapamycin for 6 months highlighted some degree of reduction in the volumes of sclerotic plaques, accompanied by a significant decrease in T-responder cells [160]. To the best of our knowledge, there is no specific agent that can modulate the selective autophagic removal of mitochondria during neurodegeneration. Several efforts are ongoing to try to overcome this shortcoming. For example, mitophagy may be improved by small molecules activating the PINK1–Parkin pathway. The ATP analog kinetin triphosphate has been identified as a potent PINK1 activator. Indeed, this neosubstrate accelerates PINK1-dependent Parkin recruitment to damaged mitochondria and prevents the apoptosis induced by oxidative stress in neuronal cells [161]. Unfortunately, long-term oral kinetin administration does not prevent the neurodegeneration induced by α-syn in a PD model [162]. A recent high-throughput screening identified two other small molecules (T0466 and T0467) that affect the PINK1–Parkin axis. These compounds successfully promote Parkin translocation to mitochondria, suppress mitochondrial aggregation in dopaminergic neurons, and improve locomotor defects in the *Drosophila* PINK1 model [163]. Finally, it has been suggested that Rho-associated protein kinase (ROCK) inhibitors may exert a neuroprotective effect by increasing the activity of the Parkin-mediated mitophagy pathway. In one investigation, the authors performed a screen of ~3000 compounds with the aim of identifying compounds that promote Parkin translocation to mitochondria. As a result, they found that several ROCK inhibitors increased the recruitment of Parkin to damaged mitochondria and found compound SR3677 to be the most efficacious. In addition, SR3677 also exerted neuroprotective effects and restored locomotor abilities in a *Drosophila* PD model [164].

The activation of autophagy seems to be efficacious in increasing the clearance of protein aggregates. Autophagic activation may be obtained by using rapamycinRAPAMYCIN and its analogs, such as TEMSIROLIMUS. These compounds permit inhibiting the mammalian target of rapamycin (mTOR). A similar effect was obtained by using the pro-neurogenic and antihistaminic compound LATREPIRIDINE. Autophagy may also be activated by potentiating the activity of 5′ adenosine monophosphate-activated protein kinase (AMPK) with RESVERATROL and a series of small molecules (A769662, GSK621, RSVA314, and RSVA405) recently identified. Autophagy activation may also be obtained in an mTOR/AMPK-independent manner. LITHIUM was found to block the pro-autophagic function of inositol monophosphatase (IMPase), and two molecules (AUTEN-69 and -99) antagonize the autophagic inhibitor phosphatase MTMR14. Oppositely, autophagy inhibition represents a possible therapeutic approach against multiple sclerosis (MS). In particular, diverse studies suggest that the anti-autophagic compound chloroquine (CQ) suppresses inflammation and reduces the clinical score of an MS mouse model. 

## 8. Conclusions and Future Perspectives

In this review, we summarize studies showing how autophagy and mitophagy have crucial roles in neurodegenerative disorders. It is clear that compromised autophagy and mitophagy dynamics mediate the pathogenesis and progression of diseases characterized by the uncontrolled accumulation of protein aggregates, such as AD and PD. These mechanisms are in contrast with those in MS, in which the neurodegenerative condition is not due to aberrant protein inclusions and autophagy and mitophagy appear to be excessively activated and to provoke deleterious conditions causing cell death or the impairment of normal cellular functions.

Interestingly, these scenarios are present when autophagy markers or any biochemical or molecular markers are analyzed in body fluids from persons affected by specific neurodegenerative conditions. Indeed, most of the investigations performed highlight a reduced activity of the autophagy processes in AD and PD; however, these processes are increased in human MS samples. Furthermore, autophagy levels correlate with clinical outcomes. Overall, these findings clearly show that changes in autophagy and mitophagy elements may be reliable markers for predicting/controlling disease progression and helping to monitor clinical status. It is, thus, necessary to point out that most of the investigations performed have utilized serum. This biofluid is more accessible and safer and has fewer contraindications than CSF, and it would certainly facilitate the continuous tracking of disease progression. Another point should be highlighted: the findings presented herein also suggest that the modulation of autophagy dynamics with pharmacological strategies may be an effective method for slowing down the progression of some neurodegenerative diseases. Different studies performed in cellular and animal models support this possibility. For instance, the inhibition of autophagy appears to reduce inflammation and clinical signs in MS models, while its activation reduces protein aggregation and consequent motor and learning behavior defects in AD and PD models. Consistent with these findings, long-term treatment with the autophagy inducer lithium in a clinical trial restored multiple cognitive parameters in AD patients.

However, it should be noted that several caveats exist in the studies mentioned above. Currently, there are no good methods for accurately quantifying autophagy and mitophagy levels in samples obtained from human patients. For this reason, it is difficult to understand whether increases or decreases in autophagy levels are not due to the impairment of the correct autophagic flux. Experiments with inhibitors of the autophagosomal fusion with the lysosomal membrane may help to overcome this limitation and may be performed in human tissues, such as skin and muscle biopsies and postmortem brain tissues. However, their application in more accessible human samples, such as sera and CSF, seems very hard to achieve. Similarly, today, it remains difficult to perform real-time monitoring of all the autophagic dynamics, from autophagosomal vesicle formation to degradation, in a patient.

Additionally, it is difficult to ensure that pharmacological agents modulating autophagy in cultured cells and animal models also do so in vivo in patients affected by a neurodegenerative condition. Furthermore, drugs activating/inhibiting autophagy may exert several other effects and modulate gene expression and protein, lipid, and nucleotide synthesis. In addition, to date, no effective treatment aimed at selectively modulating only the mitophagic pathway has been tested. Finally, at first glance, serum may represent a valid alternative to CSF for monitoring clinical status. However, outside of the CSF, the concentrations of markers related to the CNS are often low, and circulating antibodies and proteases may alter the effective concentrations of proteins in peripheral tissues [165].

In summary, more work is required to fully clarify all the connections that exist between autophagy processes and neurodegenerative status. Despite this limitation, the current literature demonstrates that the recent decades have been characterized by a great improvement in the understanding of the connection between autophagic dynamics and neurodegeneration. Currently, autophagy is accepted worldwide to be a fundamental aspect of the onset and progression of almost all neurodegenerative diseases. A better understanding of the molecular partners participating in this cellular process may help to identify susceptible patients, control disease progression, and perform active monitoring of treatment responses. Finally, the pharmacological modulation of autophagy may represent an attractive tool for identifying new disease-modifying therapies to combat different types of neurodegenerative diseases.

## Figures and Tables

**Figure 1 biomedicines-09-00149-f001:**
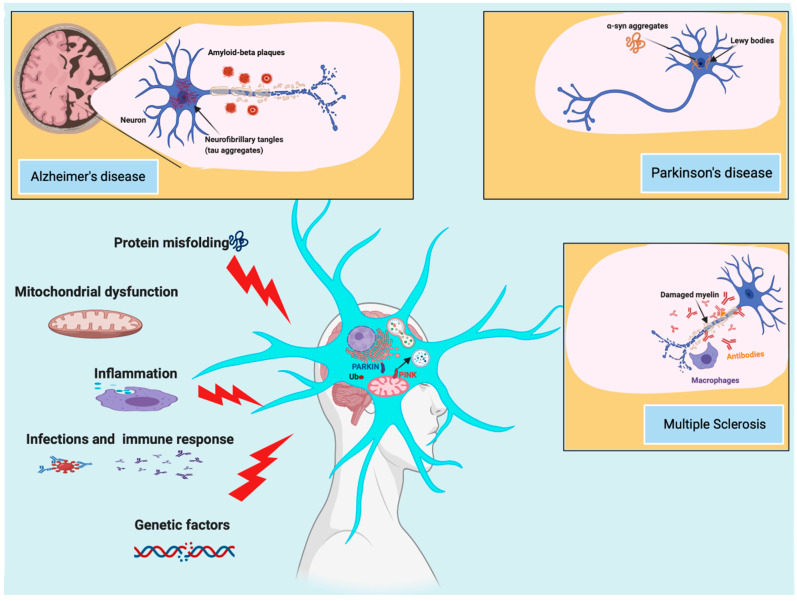
Causes and risk factors of Alzheimer’s disease (AD), Parkinson’s disease (PD), and multiple sclerosis (MS). AD and PD are the most common neurodegenerative diseases, characterized by an abnormal aggregation of protein inclusions in the brain. While in AD, the progressive loss of neurons is caused by an aberrant accumulation of beta-amyloid and tau protein, PD displays inclusions named Lewy bodies, where α-synuclein (α-syn) is the major constituent. MS is the most frequent cause of disability among young adults after traumatic brain injury. The neurodegenerative condition of MS is not due to an excessive accumulation of misfolded protein but to adverse immune-mediated processes and chronic inflammation that provoke demyelinating and neurodegenerative processes during the entire life of the patient. Among the different molecular mechanisms and risk factors involved in these neurodegenerative conditions, it has been demonstrated that autophagy and mitophagy play an important role.

**Figure 2 biomedicines-09-00149-f002:**
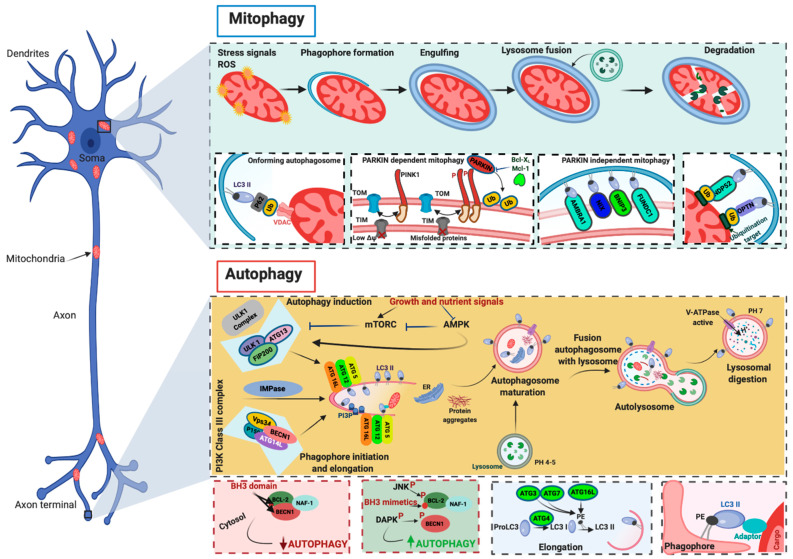
Molecular mechanisms of autophagy and mitophagy. The mammalian target of rapamycin (mTOR) and the 5′ adenosine monophosphate-activated protein kinase (AMPK) are the main negative and positive regulators of autophagy, respectively. One of the primary targets of the action of mTOR and AMPK is the unc-51-like autophagy-activating kinase 1 (ULK1)/autophagy-related (ATG) 13/FIP200 (200-kDa focal adhesion kinase family-interacting protein) complex, which is the main regulator of autophagosomal formation. Other important proteins that participate in this molecular process are the coiled-coil, moesin-like BCL-2 interacting protein (BECN1), class III phosphatidylinositol 3-kinase (PI3K), vacuolar protein sorting 34 (Vsp34), ATG14L, p150, and IMPase. The activity of BECN1 in regulating the autophagy process is also mediated by the interaction with BCL-2. During the elongation of the autophagosome, a series of autophagy-related (ATG) proteins are involved. In particular, two specific complexes were found to be essential for completing autophagosomal formation: (ATG)12–ATG5–ATG16L1 and microtubule-associated protein 1A/1B-light chain 3 (LC3)–phosphatidylethanolamine (PE) complexes. Mitochondria are particularly vulnerable to stress signals, such as ROS, which, in turn, can cause severe mitochondrial dysfunction and activate the mitophagic process. PINK1 senses this mitochondrial damage and phosphorylates and recruits Parkin to the outer mitochondrial membrane of the mitochondria. Phosphorylation converts Parkin to an active ubiquitin (Ub)-dependent enzyme and mediates the phosphorylation of different mitochondrial proteins. During this process intervene different Ub-binding autophagy receptors such as p6, NBR1, NDP52, and optineurin (OPTN), which connect the damaged mitochondria to the forming autophagosomes. Mitophagy may also be executed in a Parkin-independent manner. In this case, different proteins (FUNDC1, AMBRA1, NIX, and BNIP3) intervene to signal the mitochondria that should be degraded.

**Figure 3 biomedicines-09-00149-f003:**
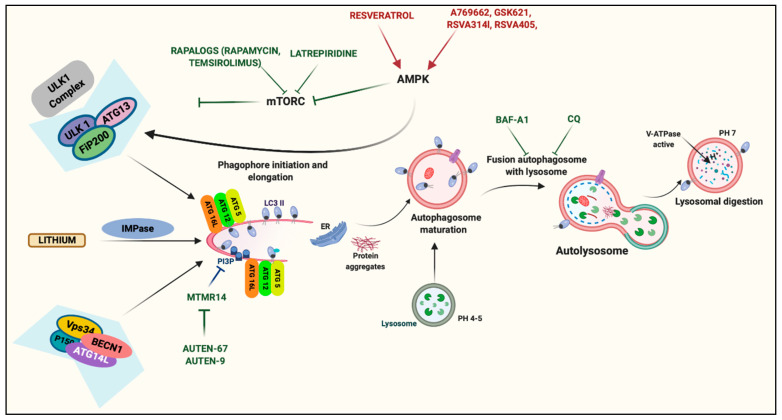
Strategies for targeting autophagy in neurodegeneration.

**Table 1 biomedicines-09-00149-t001:** Summary of autophagy- and mitophagy-related markers in biofluids of MS-, AD-, and PD-affected persons.

Neurodegenerative Condition	Marker	Role	Type of Human Biofluid
**MS**	Parkin	Mitophagy regulator	Serum, CSF
	ATG5	Autophagy regulator	Serum, CSF
	Mitochondrial adenine nucleotide translocase 1 (ANT1)	Mitochondrial ADP/ATP translocase	Serum, CSF
	Translocator protein 18 kDa (TSPO)	Regulator of mPTP opening	Blood PBMCs
	Growth/differentiation factor 15 (GDF-15)	Mitochondrial disease marker	Serum
	TNFα	Proinflammatory cytokine	Serum, CSF
	Lactate	Mitochondrial dysfunction marker	Serum, CSF
**AD**	BECN1	Autophagy regulator	Blood PBMCs, serum
	p62	Autophagy regulator	Blood PBMCs
	LC3	Autophagy regulator	Blood PBMCs
	ATG5	Autophagy regulator	Plasma, serum
	Parkin	Mitophagy regulator	Serum
	EEA1, LAMP1, LAMP2, RAB3, and RAB7	Lysosomal regulators	CSF
**PD**	LC3B	Autophagy regulator	CSF
	BECN1	Autophagy regulator	CSF, blood PBMCs
	ATG5	Autophagy regulator	CSF
	LAMP2	Lysosomal regulator	CSF
	ULK1	Autophagy regulator	Blood PBMCs
	ATG5	Autophagy regulator	Blood PBMCs
	ATG4B	Autophagy regulator	Blood PBMCs
	ATG16L1	Autophagy regulator	Blood PBMCs

## Data Availability

Not applicable.

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
