# Peer review of "Relevance of Autophagy and Mitophagy Dynamics and Markers in Neurodegenerative Diseases"

_biomedicines, 2021, doi:10.3390/biomedicines9020149_

Round 1

Reviewer 1 Report

The manuscript by Giorgi et al. provides a review of the current literature about the role of mitophagy and autophagy in neurodegenerative diseases with a focus on Alzheimer, Parkinson, and multiple sclerosis. The review is well written and the paragraphs nicely organized. Also, the figures are good. I would suggest minor changes before publication:

-) at page 3, legend of figure 1, 4th row, I would replace “meanwhile” at the beginning of the sentence with “while”.

-) at page 3, row 22, I would include a recent report (Sato et al., 2019, J Pharm Science) describing the role of rapamycin during mammalian microautophagy, in the paragraph describing microautophagy.

-) at page 4, rows 27-31, I would include some recent works (Chang et al., 2020, Autophagy, in drosophila; Aspernig et al., 2019, Cell Rep, in C. elegans) demonstrating that mTORC2 can regulate autophagy, too.

-) at page 4, rows 32-38, I would spend more words to better explain in more details the different phosphorylation sites of ULK1, how they are regulated by different kinases (mTOR and AMPK) and how they regulate in turn autophagy.

-) at page 8, rows 23-24, in addition to reference 61, I would add another recent report demonstrating the modulation of ATG5 in naïve-to-treatment MS patients (Paunovic et al., 2018, J Neuroimmunology).

-) at page 8, rows 23-38, I would add some recent works describing the role of another autophagy protein, i.e. Ambra1, in the regulation of Treg cells also in a EAE model (Becher et al., 2018, Dev Cell; Akatsuka et al., 2017, Bioch Bioph Res Commun).

-) the paragraph 5 (from page 8, row 27) lists a number of alterations observed in autophagy proteins in AD patients. It could be good if the authors, at the end of the paragraph, could comment on the applicability of these alterations as biomarkers in AD. How could we use all these data to choose some good biomarker?

-) the paragraph 6 (from page 9, row 47) could include a recent report indicating a role of endosomal-microautophagy in PD (Caballero et al., 2018, Aging Cell). Also, I would conclude paragraph 6 (but also the number 5 and 4) with few rows summarizing all the described data, just to give readers some take-home message.

Author Response

Reply to Reviewers

Author's Reply to the Review Report (Reviewer 1)

The manuscript by Giorgi et al. provides a review of the current literature about the role of mitophagy and autophagy in neurodegenerative diseases with a focus on Alzheimer, Parkinson, and multiple sclerosis. The review is well written and the paragraphs nicely organized. Also, the figures are good. I would suggest minor changes before publication:

  • at page 3, legend of figure 1, 4throw, I would replace “meanwhile” at the beginning of the sentence with “while”.

We thank the reviewer for the suggestion. We replace “meanwhile” with “while”

  • at page 3, row 22, I would include a recent report (Sato et al., 2019, J Pharm Science) describing the role of rapamycin during mammalian microautophagy, in the paragraph describing microautophagy.

We thank the reviewer for the suggestion. We not only inserted the indicated manuscript, but we also improved the section regarding microautophagy

  • at page 4, rows 27-31, I would include some recent works (Chang et al., 2020, Autophagy, in drosophila; Aspernig et al., 2019, Cell Rep, in C. elegans) demonstrating that mTORC2 can regulate autophagy, too.

We thank the reviewer for the recommendations. Now, we have included recent works demonstrating that mTORC2 also regulate the autophagic and mitophagic processes.

  • at page 4, rows 32-38, I would spend more words to better explain in more details the different phosphorylation sites of ULK1, how they are regulated by different kinases (mTOR and AMPK) and how they regulate in turn autophagy.

We thank the reviewer for the recommendations. We have extend the section regarding the phosphorylation sites of ULK1, how these sites are target for mTOR and AMPK and how these modifications regulate autophagy.

  • at page 8, rows 23-24, in addition to reference 61, I would add another recent report demonstrating the modulation of ATG5 in naïve-to-treatment MS patients (Paunovic et al., 2018, J W

We thank the reviewer for the comment. We have included the suggested reference and a brief sentence reporting the main findings of the work of Paunovic et al.

  • at page 8, rows 23-38, I would add some recent works describing the role of another autophagy protein, i.e. Ambra1, in the regulation of Treg cells also in a EAE model (Becher et al., 2018, Dev Cell; Akatsuka et al., 2017, Bioch Bioph Res Commun).

We thank the reviewer for the comment. Now, we added the important contribution of AMBRA1 for Treg cells.

  • the paragraph 5 (from page 8, row 27) lists a number of alterations observed in autophagy proteins in AD patients. It could be good if the authors, at the end of the paragraph, could comment on the applicability of these alterations as biomarkers in AD. How could we use all these data to choose some good biomarker?

We thank the reviewer for the suggestion. We added some sentence at the end of the paragraph where we commented the importance of autophagy and mitophagy in AD and discussed about the possibility to use their circulating element as new biomarker.

  • the paragraph 6 (from page 9, row 47) could include a recent report indicating a role of endosomal-microautophagy in PD (Caballero et al., 2018, Aging Cell). Also, I would conclude paragraph 6 (but also the number 5 and 4) with few rows summarizing all the described data, just to give readers some take-home message.

We thank the reviewer for the recommendation. We included and discussed the important work of Caballero et al. in the PD section and also included summaries at the end of paragraph 4, 5 and 6.

Reviewer 2 Report

The paper by Giorgi et al is well written and comprehensive review of the involvement of mitophagy in neurodegenerative disease pathogenesis. The language is good, the text scientifically sound and all important aspects are well covered. I only have few very minor comments.

Page 2 row 3-4 The sentence seems to imply that a-syn mutations are the only cause for a-syn aggregation, however this is not true as as aggregation is seen also in pd patients with no SNCA mutations.

Figure 1 AD part, in the image the neurofibrillary tangles within the neurons are depicted separately from the tau aggregates that are depicted as extracellular, shouldn’t the tau aggregates form the intracellular tangles?

Page 15 rows 29-32. Indeed, the authors are completely right that to analyze autophagy the flux should be studied and not just the steady state levels of the autophagy related proteins. But what do the authors mean by difficult to control autophagic flux? The flux can be easily controlled by blocking the fusion of autophagosomes with lysosomes for example by chloroquine. This allows analysis of the newly formed autophagosomes separately from the lysosomal breakdown and analysis of both blocked and unblocked states reveals the rate of the autophagic flux.

Author Response

Author's Reply to the Review Report (Reviewer 2)

The paper by Giorgi et al is well written and comprehensive review of the involvement of mitophagy in neurodegenerative disease pathogenesis. The language is good, the text scientifically sound and all important aspects are well covered. I only have few very minor comments.

  • Page 2 row 3-4 The sentence seems to imply that a-syn mutations are the only cause for a-syn aggregation, however this is not true as as aggregation is seen also in pd patients with no SNCA mutations.

We thank the reviewer for the comment. We have corrected the sentence specifying that not only SNCA mutation determine a-syn aggregation.

  • Figure 1 AD part, in the image the neurofibrillary tangles within the neurons are depicted separately from the tau aggregates that are depicted as extracellular, shouldn’t the tau aggregates form the intracellular tangles?

We thank the reviewer for the comment and we apologize for the mistake. We corrected the figure. Now, it is clear that tau aggregates form the intracellular tangles.

  • Page 15 rows 29-32. Indeed, the authors are completely right that to analyze autophagy the flux should be studied and not just the steady state levels of the autophagy related proteins. But what do the authors mean by difficult to control autophagic flux? The flux can be easily controlled by blocking the fusion of autophagosomes with lysosomes for example by chloroquine. This allows analysis of the newly formed autophagosomes separately from the lysosomal breakdown and analysis of both blocked and unblocked states reveals the rate of the autophagic flux.

We thank the reviewer for the comment. We agree that the autophagic flux may be easily controlled by using inhibitors of autophagic flux. Thus, we adjusted the indicated sentence and we reported that it is true that the effective autophagic dynamics can be controlled in vitro in human samples (such as brain tissue), but it is hard to monitor in other human samples, such as human bio fluids.

Reviewer 3 Report

No comment for the authors. In my opinion, the paper is original, well structured and original. It provides useful and new information on the approach to neurodegenerative diseases. English language and style are fin
The article can be published in its current form

Thanks

Author Response

Author's Reply to the Review Report (Reviewer 3

  • No comment for the authors. In my opinion, the paper is original, well structured and original. It provides useful and new information on the approach to neurodegenerative diseases. English language and style are fine. The article can be published in its current form. Thanks

We thank the reviewer to have appreciated our efforts.